# SARS-CoV-2: An Updated Review Highlighting Its Evolution and Treatments

**DOI:** 10.3390/vaccines10122145

**Published:** 2022-12-14

**Authors:** Xirui Zhang, Hao Yuan, Zipeng Yang, Xiaoyu Hu, Yasser S. Mahmmod, Xiaojing Zhu, Cuiping Zhao, Jingbo Zhai, Xiu-Xiang Zhang, Shengjun Luo, Xiao-Hu Wang, Mengzhou Xue, Chunfu Zheng, Zi-Guo Yuan

**Affiliations:** 1Guangdong Provincial Key Laboratory of Zoonosis Prevention and Control, College of Veterinary Medicine, South China Agricultural University, Guangzhou 510642, China; 2Institute of Animal Health, Guangdong Academy of Agricultural Sciences, Key Laboratory of Livestock Disease Prevention of Guangdong Province, Scientific Observation and Experiment Station of Veterinary Drugs and Diagnostic Techniques of Guangdong Province, Ministry of Agriculture and Rural Affairs, Guangzhou 510640, China; 3Infectious Diseases, Department of Animal Medicine, Faculty of Veterinary Medicine, Zagazig University, Zagazig 44511, Egypt; 4Veterinary Sciences Division, Al Ain Men’s College, Higher Colleges of Technology, Abu Dhabi 17155, United Arab Emirates; 5The 80th Army Hospital of the Chinese people’s Liberation Army, Weifang 261021, China; 6Key Laboratory of Zoonose Prevention and Control at Universities of Inner Mongolia Autonomous Region, Medical College, Inner Mongolia Minzu University, Tongliao 028000, China; 7Department of Cerebrovascular Diseases, The Second Affiliated Hospital of Zhengzhou University, Zhengzhou 450014, China; 8Department of Microbiology, Immunology and Infectious Diseases, University of Calgary, Calgary, AB T2N 4N1, Canada

**Keywords:** SARS-CoV-2, COVID-19, evolution, vaccines, drug, treatments

## Abstract

Since the SARS-CoV-2 outbreak, pharmaceutical companies and researchers worldwide have worked hard to develop vaccines and drugs to end the SARS-CoV-2 pandemic. The potential pathogen responsible for Coronavirus Disease 2019 (COVID-19), SARS-CoV-2, belongs to a novel lineage of beta coronaviruses in the subgenus arbovirus. Antiviral drugs, convalescent plasma, monoclonal antibodies, and vaccines are effective treatments for SARS-CoV-2 and are beneficial in preventing infection. Numerous studies have already been conducted using the genome sequence of SARS-CoV-2 in comparison with that of other SARS-like viruses, and numerous treatments/prevention measures are currently undergoing or have already undergone clinical trials. We summarize these studies in depth in the hopes of highlighting some key details that will help us to better understand the viral origin, epidemiology, and treatments of the virus.

## 1. Introduction

*Coronaviridae* is a single-strand, positive-sense, enveloped RNA virus family [1], circulating in many avian and mammal species hosts [2]. In December 2019, a patient was found to have pneumonia caused by an unknown betacoronavirus. With the unbiased sample sequencing of the patient, a novel coronavirus was identified and named Severe Acute Respiratory Syndrome Coronavirus 2, SARS-CoV-2, by the International Committee on Taxonomy of Viruses (ICTV).

SARS-CoV-2 is a beta coronavirus belonging to the subgenus baculovirus of the *Coronaviridae* family (Beta-CoV lineage B) based on a phylogenetic analysis [3]. Similar to other coronaviruses, it is a positive-sense single-stranded RNA (+ssRNA) virus with an envelope, and its virions are round or oval with a diameter of 50 to 200 nm [4]. Typical betacoronavirus structures (S, E, and M genes) were identified [5]. Notably, the spike (S) protein is one of the major viral proteins, always used for viral typing [5]. The S protein could be divided into the S1 and S2 domains. Each domain is responsible for different functions, where the S1 domain is responsible for receptor binding, and the S2 domain is responsible for cell membrane fusion [6]. The S protein is also critical in determining the host’s transmission and tropism [7].

The virus can spread from person to person, which has been confirmed [8]. The sources of infection seen so far are mainly other patients with COVID-19 infection. Asymptomatic patients can also be contagious [8]. Respiratory droplets and close contact are the two main methods of transmission. There is potential for aerosol transmission when exposed to high aerosol concentrations for a long time in a reasonably confined environment. The likelihood of a virus spreading increases in a crowded or cramped environment [8].

The viral basic reproduction number (R0) has been estimated to be 1.4–3.9, with an incubation period of 1–14 days and an average of 4–8 days, typically 3–7 days [9,10,11]. The fatality rate is around 2%, with fever, dry cough, and fatigue as the main manifestations. Its viral diagnostic method had already been developed based on PCR and RT-PCR [12,13,14,15]. Since the E gene and the RdRp gene are comparatively more sensitive, they were chosen for further evaluation for detection [13]. A successful vaccination protocol is urgently needed to combat the SARS-CoV-2 outbreak [16].

We review the extensive studies that have already been conducted based on the genome sequence analysis of SARS-CoV-2 and its comparison with that of other SARS-like viruses to highlight some key aspects and provide answers to some of the most important questions regarding its origin, epidemiology, and treatment to better combat the virus. Based on this, useful insights for creating a quick and precise viral diagnostic approach will also be provided.

## 2. SARS-CoV-2’s Origin

Although the natural viral reservoir is believed to be bats, with 96.2% sequence identity [17], its intermediate host has yet to be determined.

An experiment carried out by researchers from HK also displayed 85.5–92.4% sequence identity with that of the pangolin-derived coronavirus, together with some other evidence suggesting that Malayan pangolins in Southern China could be an intermediate host [18].

Other than pangolins, snakes and minks were suspected to be intermediate hosts for a short time. However, after an intense debate, all proposals were rejected. The most debated SARS-CoV-2 intermediate hosts can be found in Biorxiv (accessed on 8 September 2020). Discussion of which animal is the intermediate host has resulted in no conclusive answer, leaving the question unresolved. The only factor that has been determined is that the virus originates from bats [17].

## 3. Evolutionary Findings Relating to the Novel Coronavirus

To find out more about its relationship with SARS and MERS, Lu’s research group has reported that SARS-CoV-2 shares 79% sequence identity with SARS-CoV. However, this is a lower level of identity than the one of MERS-CoV (with only 50%) [19,20]. Classification by ORF1ab of seven conserved replicase domains also indicated that SARS-CoV-2 and SARS-CoV belong to the same species, with 94.6% amino acid identity [21].

Meanwhile, a previous study showed that it formed an isolated clade composed of two other virus strains originating from bats, ZC45 and ZXC21 [22]. The other study showed that SARS-CoV-2 had formed a considerably long branch length with bat-SL-CoVZC45 and bat-SL-CoVZXC21 in a phylogenetic tree distinct from SARS [19]. In another study, samples from Wuhan patients were examined and shared 88% sequence identify with bat-SL-CoVZC45 and bat-SL-CoVZXC21 [19]. Notably, another two studies reported that the viral genome had approximately 89% [23] and 86.9% [14] nucleotide sequence identity with that of the bat SARS-like CoV (bat-SL-CoVZC45) genome. However, another study claimed that a bat-originated coronavirus termed RaTG13 is the closest relative of the virus from Wuhan, based on the RdRp and S genes’ sequences [24].

Phylogenetic tree analysis employing Maximum Likelihood (ML) methods by Benvenuto et al. had shown the same outcome, with the SARS virus, SARS-like bat-derived virus and the novel CoV together forming one clade (termed clade II) and the MERS viruses themselves forming another (termed clade I) [25]. Within clade II, the bat-derived SARS-like coronavirus and SARS-CoV-2 formed the IIa cluster, and the bat-derived SARS-like coronavirus and the SARS virus formed the IIb cluster.

The next three-dimensional modeling verification further demonstrates that the SARS-CoV-2 receptor binding domain is closer to the one from SARS-CoV, despite the SARS-CoV-2 complete genome being reported to be closer to that of bat-SL-CoVZC45 and bat-SL-coVZXC21 [26].

## 4. Clinical Manifestations and Pathological Features

Fever, dry cough, and fatigue are the main manifestations. A few people experience diarrhea, myalgia, sore throat, and nasal congestion [27]. Severe patients typically have dyspnea or hypoxemia one week after the symptoms start. They can quickly progress to severe metabolic acidosis, septic shock, coagulation malfunction, acute respiratory distress syndrome, and failure of several organs [28].

It is important to note that severely and critically unwell individuals may exhibit moderate, mild, or no overt fever. In contrast, people with mild symptoms will exhibit a low fever and slight fatigue, instead of any pneumonia symptoms. Hence, only a minority of the patients in the present cases would develop a critical condition, giving the majority of patients a decent outlook. The death rate for COVID-19 in older patients may be higher. Older people and those with persistent underlying conditions have a dismal prognosis, and the most prevalent clinical signs are fever, dry cough, and malaise. A few people experience diarrhea, myalgia, conjunctivitis, runny nose, and nasal congestion [29]. In addition, elderly patients aged 65 and older are more likely to develop COVID-19 infection sequelae, including hypertension, diabetes, chronic obstructive pulmonary disease, and dysphrenia [30].

Children often experience minor symptoms within a week, with mild to moderate fever as the most common presentation [31]. A few children can experience high fever, dry cough, and other respiratory symptoms. Severe cases can cause hypoxia, multi-organ failure, or acute respiratory distress syndrome (ARDS) [32,33].

In addition, atypical skin symptoms from COVID-19 infection include a maculopapular rash, urticaria, pseudo-chilblain, and blisters [34]. Hyposmia and anosmia is also a symptom of COVID-19 infection. The viral invasion into olfactory epithelium cells might result in the temporary malfunction of the cells, hyposmia, and anosmia. However, the symptoms are only temporary, and most will subside within a week or a few weeks [35]. The pathological features of COVID-19 are very similar to those of SARS and MERS infections [36]. Additionally, liver tissue demonstrates modest lobular active inflammation and significant microvascular septoplasty. However, neither drug-induced liver damage nor infection with the SARS-CoV-2 virus is backed by solid evidence. The cardiac tissue exhibited no overt histological alterations, indicating that the SARS-CoV-2 infection might not directly harm the heart. Lymphopenia is a prevalent symptom in COVID-19 patients.

Moreover, another study found that severe patients often have neurological symptoms, compared with non-severe COVID-19 patients, with acute cerebrovascular disease, disturbance of consciousness, and skeletal muscle symptoms [37]. After analyzing 522 samples from SARS-CoV-2 patients, Diao et al. [38] discovered that SARS-CoV-2 patients, especially elderly patients (over 60 years of age) and patients in need of intensive care unit (ICU) treatment, had significantly reduced total numbers of T cells, CD4^+^ T cells, and CD8^+^ T cells. The number of T cells was negatively correlated with serum IL-6, IL-10, and TNF-α concentrations. When the T cell count increased, the patients’ IL-6, IL-10, and TNF levels decreased. It is worth mentioning that patients with SARS-CoV-2 had significantly higher levels of PD-1, a marker of T cell depletion. In addition, PD-1 and Tim-3 from T cells increased during the shift from the precursor stage to the apparent symptomatic stage, implying a decrease in T cells and functional depletion in SARS-CoV-2 patients. Finally, a study found that patients with renal injury had a greater probability of in-hospital death by examining 710 SARS-CoV-2 patients [39].

## 5. Treatments under Development

Since the outbreak has substantially affected societies around the world, many scientists have focused on seeking a viable solution to combat the disease. To date, many drugs have been placed under development and even started clinical trials.

### 5.1. Antiviral Drugs for the Treatment of COVID-19

#### 5.1.1. Remdesivir

Remdesivir is an adenosine analog that works as an antiviral agent in cultured cells and animal models against various RNA viruses, including SARS/MERS. It was initially an experimental antiviral drug used to treat Ebola and is now under the cooperation of Gilead Sciences Inc. with researchers and clinicians in China. It can cause the RNA chains of the viral to terminate before it reaches its mature stage and therefore fulfills its antiviral function [40]. A previous study showed that Remdesivir could effectively suppress the infection of the virus in a highly sensitive human cell line (human liver cancer Huh-7 cells) [41]. From their detection, Remdesivir functions in one stage after virus entry [41]. The EC_90_ value of Remdesivir against the SARS-CoV-2 infection of E6 cells was 1.76 μM. Meanwhile, the EC_50_ value is 0.77 μM (SI  >  129.87) [41]. In America, the first SARS-CoV-2 patient was treated with Remdesivir and was already cured [42]. Moreover, in France, a patient was treated with Remdesivir and cured [43]. However, Fan et al. [44] reported that administrating Remdesivir to mice with a daily dose of 150 μg/mice can cause a significant decrease in sperm count and motility and a significant increase in abnormal sperm rate, indicating that the administration of Remdesivir may cause testicular toxicity. Since the experiment was only carried out on mice, more experiments are needed to confirm the result [44]. Moreover, on April 29, 2020, the results of three clinical trials of Remdesivir for COVID-19 were announced simultaneously. These studies reported contradicting results on the efficacy of Remdesivir in treating COVID-19. The first report was from the first clinical trial to evaluate Remdesivir in the treatment of COVID-19, initiated by NIH [45]. In this randomized controlled trial with 1063 patients, preliminary data analysis revealed that advanced COVID-19 patients who received Remdesivir recovered more quickly than comparable patients who received a placebo. Another study evaluated the 5-day and 10-day treatment courses of Remdesivir for COVID-19 patients and found no significant difference (no significant adverse effect), and all patients’ clinical conditions showed a similar improvement [46]. However, the third study presented a conclusion that conflicted with the findings of the first two research works. It was a randomized, double-blinded, placebo-controlled study to evaluate Remdesivir’s effectiveness in treating patients with severe COVID-19 [47]. Their result showed no statistically significant difference in clinical improvement time between the group receiving Remdesivir and the group receiving a placebo. The result is surprising, as the two previously mentioned trials had at least shown some benefits in treating the disease. According to a study, Remdesivir treatment reduces mortality in SARS-CoV-2-infected individuals. In this study of 342 patients, Raltegravir was given to 60 patients in the control group and 18 in the case group (35.1% vs. 10.5%, *p* < 0.0001). Remdesivir therapy can decrease SARS-CoV-2 mortality [48]. A randomized, double-blind, placebo-controlled trial validated the safety of Remdesivir in a three-day treatment [49]. Some studies have confirmed that a three-day outpatient course of Remdesivir significantly reduces the risk of hospitalization or death. In a retrospective matched-pair study to evaluate the contributions of Remdesivir and vaccination in reducing severe clinical outcomes, patients who received a three-day course of Remdesivir after receiving ambulatory vaccination were 75% less likely to end up in the hospital and 95% less likely to experience respiratory failure. Neither intubations nor fatalities occurred [50].

#### 5.1.2. Arbidol Hydrochloride

A broad-spectrum antiviral drug called arbidol hydrochloride prevents enveloped viruses from infecting host cells by preventing the fusion of the virus with the cell membrane. Including FLU-A, RSV, HRV 14, and CVB3, Arbidol exhibits strong inhibitory efficacy against both enclosed and non-enveloped RNA viruses. According to the records of 252 COVID-19 patients, patients who received Arbidol experienced a clinical improvement rate that was noticeably higher than that of patients who did not. Clinical improvement rates in the Arbidol group were 95.6% and 81.7%, respectively, in moderately and severely ill patients, which was considerably greater than in the no-Arbidol group (66.6% and 53.8%). However, there was no discernible difference among critically ill individuals. Additionally, results show that Arbidol works well in treating COVID-19 patients and may be a useful antiviral therapeutic option for those with mild to moderate COVID-19 symptoms [51].

#### 5.1.3. Favipiravir

A powerful inhibitor of viral RNA polymerase, favipiravir is phosphor-ribosylated by cellular enzymes to produce favipiravir-ribofuranosy l-5′-triphosphate, which is its active form (RTP). When used to treat influenza, the guanine analog favipiravir effectively inhibits RNA viruses’ RNA-dependent RNA polymerase. With an IC50 of 341 nM, favipiravir-RTP blocks influenza viral RNA-dependent RNA polymerase (RdRP) activity [52].

A recent study reported its activity in SARS-CoV-2 [53]. A randomized trial of favravavir in combination with interferon-α (CHICTR090029) and barbituric (CHICTR09009554) for SARS-CoV-2 has been conducted [54].

#### 5.1.4. Baricitinib

The medication baricitinib particularly inhibits Janus-activated kinase 1 and 2 (JAK1/2), which mediate signaling for cytokines and growth factors involved in hematopoiesis, inflammation, and the immune response. Suppressing JAK1 and JAK2’s enzymatic activity could modify intracellular signaling by reducing STAT phosphorylation and activation. An observational, long-term experiment involved treating patients with 4 mg of baricitinib once a day for 7 days after receiving 4 mg twice daily for 2 days. Serum-derived cytokine and antibodies against SARS-CoV-2 were assessed and linked with changes in the immunological phenotype and phosphorylated STAT3 (p-STAT3) expression in blood cells (anti-SARS-CoV-2) [55]. A single treated patient with altered myeloid cell functional activity was assessed. According to the findings, patients using baricitinib experienced significantly lower levels of IL-6, IL-1, and TNF in their serum, rapid recovery in the frequency of circulating T and B cells, and increased antibody production against the SARS-CoV-2 spike protein. By altering the patients’ immunological environment, baricitinib prevented the viral infection from progressing to a severe, extreme form, and these modifications were linked to a safer, better clinical result for COVID-19 pneumonia patients. Recently, baricitinib (Olumiant) received U.S. Food and Drug Administration approval as the first immunomodulatory treatment for COVID-19 [56].

#### 5.1.5. Tofacitinib

The JAK family of kinases is effectively and selectively inhibited by tofacitinib. Tyrosine-protein 2 kinases (Tyk2) and JAK1, JAK2, and JAK3 are less activated when treated with tofacitinib [57]. Tofacitinib blocks the signaling of hetero-dimeric cytokine receptors in human cells, which have higher functional selectivity to bind JAK3 and/or JAK1 but not homodimer JAK2. Tofacitinib reduces interleukin signaling, including IL-2, IL-4, IL-6, IL-7, IL-9, IL-15, and IL-21, and interferon type I and II, and modifies immunological responses. A total of 289 patients were randomly assigned at 15 sites around Brazil [34]. In particular, 89.3% of the patients received glucocorticoids during hospitalization. By day 28, 18.1% of deaths or respiratory failures were noted in the tofacitinib group and 29.0% in the placebo group. Moreover, by day 28, 2.8% of patients in the tofacitinib group and 5.5% of patients in the placebo group had died from any cause. When comparing tofacitinib to a placebo, the proportional chances of having a lower score on the eight-level ordinal scale were 0.60 on day 14 and 0.54 on day 28. Twenty patients (14.1%) using tofacitinib and 17 (12.0%) taking a placebo experienced serious adverse events. By day 28, tofacitinib resulted in a lower risk of death or respiratory failure among patients hospitalized with COVID-19 pneumonia compared to those treated with a placebo [58].

#### 5.1.6. Molnupiravir

The ribonucleoside analog EIDD-1931 has a prodrug called molnupiravir that is orally bioavailable. Several coronaviruses, including SARS-CoV-2, MERS-CoV, and SARS-CoV, are resistant to the antiviral effects of molnupiravir, which has a broad spectrum of antiviral activity [59]. The potential of molnupiravir was assessed in a study of COVID-19 and seasonal and pandemic influenza. In contrast to the placebo group (16.7%), viruses could only be isolated from a few of the 202 patients taking 800 mg molnupiravir (1.9%) on day 3 (*p* = 0.02). Compared to 11.1% of participants taking a placebo on day 5, viruses could not be isolated from persons taking 400 or 800 mg of molnupiravir (*p* = 0.03). Overall, molnupiravir was well tolerated, and the side effects were consistent across all groups [60]. In a propensity score matching analysis, the risk of the composite outcome was non-significantly lower when using molnupiravir: the hazard ratio was 0.83 (95% confidence interval, 0.57–1.21). Molnupiravir was correlated with a substantial reduction in the risk of a compound outcome in elderly patients, at 0.54 (0.34–0.86); in women, at 0.41 (0.22–0.77); and in patients with insufficient COVID-19 immunization, at 0.45 (0.25–0.82), according to a subgroup analysis. Statistics indicate that molnupiravir may reduce the risk of severe COVID-19 and COVID-19-related death [61]. Moreover, it was reported that complications are factors in the effectiveness of molnupiravir treatment. The mean age of the 192 patients on molnupiravir was 70.4 ± 15.4 years, and regarding comorbidities, the most prevalent were active cancer (51, 26.6%), obesity (51, 26.6%), chronic lung disease (56, 29.2%), and cardiovascular disease (96, 50.0%). Overall, 13 patients died: six died due to COVID-19, and seven died due to complications during treatment. Notably, of these patients, one had sepsis due to a urinary tract infection, two had traumatic brain injuries, one had heart failure, and four had metastatic cancer [62]. In addition, patients treated in the first three days after the onset of COVID-19 clinical symptoms have a lower risk of developing severe illness [62].

#### 5.1.7. Anakinra

Anakinra is an approved 17-kD recombinant nonglycosylated human IL-1 receptor antagonist in treating rheumatoid arthritis. By inhibiting the action of IL-1, it may be useful as an adjuvant therapeutic option in individuals with severe COVID-19. In a randomized controlled clinical trial, 30 individuals were enrolled, and 15 received anakinra. Eleven patients were female (36.7%), and 19 were male (63.3%). The patients’ average age was 55.77 + 15.89 years. Compared to the control group, the intervention group’s need for invasive mechanical ventilation was significantly lower (20.0% vs. 66.7%, *p* = 0.010). Additionally, these patients’ hospital stays were considerably shorter (*p* = 0.043). There was no discernible increase in the rate of infection. According to the findings, patients were referred to critical care units because severe COVID-19 required less mechanical ventilation when anakinra was used as an immunomodulatory drug. The drug shortened the patient’s time in the hospital. Furthermore, there was no discernible rise in the risk of infection. These results require confirmation in additional randomized placebo-controlled trials with a larger sample size [63].

#### 5.1.8. Nirmatrelvir/Ritonavir

Nirmatrelvir/ritonavir is another oral antiviral (OA) medication that helps to prevent mild to moderate COVID-19 patients from developing severe COVID-19. Nirmatrelvir is an oral protease inhibitor that can help to block the replication of the virus. Ritonavir is used as a booster for the former, slowing down the rate of decomposition and metabolism of nirmatrelvir in the liver to maintain the concentration of nirmatrelvir in the plasma and make it more effective. To date, some real-world investigations have shown that the nirmatrelvir/ritonavir combination is effective against mutant strains. It is thought to be a therapy that will alter the present outbreak prevention and control strategy. One retrospective cohort study assessed the clinical outcomes of nirmatrelvir/ritonavir in mild-to-moderate hospitalized patients [64]. Using propensity score matching, 890 recipients of nirmatrelvir and ritonavir were matched with 890 placebo recipients. When compared to controls, recipients of nirmatrelvir/ritonavir reportedly had a reduced incidence of all-cause death (*p* < 0.0001), progression of composite disease (*p* < 0.0001), and requirement for oxygen therapy (*p* = 0.032). Furthermore, a quicker reduction in viral burden in nirmatrelvir/ritonavir recipients was observed [64].

However, some are concerned that the addition of ritonavir, a CYP3A4 enzyme inhibitor, leads to interactions with various medications and impairs liver and kidney function, thereby restricting its usage among the general population and raising the risk of harm in the elderly and those with underlying conditions. In another real-world retrospective research study by Gentile et al., 111 mild to moderate patients who were older or had significant comorbidities were enrolled and received the nirmatrelvir/ritonavir combination. Despite their age and comorbidities, throughout a 14-day follow-up, only one patient was reported hospitalized. The proportion of reported ADRs was low, and no recipients stopped their therapy due to adverse drug reactions. Moreover, participants with two or more comorbidities experienced similar risks of hospitalization (1.4%) versus those with no or one comorbidity (1.6%). Although they concluded that obesity was the most prevalent comorbidity for the development of severe COVID-19, these older-aged, OA-treated recipients with comorbidities were shown to have a low hospitalization rate, mortality, and ADRs [65].

In short, oral antiviral medications nirmatrelvir/ritonavir have demonstrated significant therapeutic benefits. The World Health Organization (WHO) and National Institute of Health (NIH) have already announced on their official websites that nirmatrelvir, in combination with ritonavir, may be used in mild to moderate patients who are at high risk for developing severe COVID-19 [66].

### 5.2. Monoclonal Antibodies for the Treatment of COVID-19

#### 5.2.1. Cilgavimab and Tixagevimab

The antibody (mAb) combination AZD7442 was created with two anti-spike monoclonal antibodies: Cilgavimab (AZD1061) and Tixagevimab (AZD8895). It was claimed to be both therapeutic and suitable for pre-exposure prophylaxis. Cilgavimab and Tixagevimab are classified as receptor-binding motif (RBM) class II and class III mAbs and are capable of binding to epitopes that overlap ACE2 and limiting the motion of the spike protein, respectively [66]. Data on their effectiveness and safety in immunized individuals during the Omicron waves are currently only available from a few retrospective cohort studies.

In a trial in Israel by Kertes et al., participants were randomly assigned to receive one dosage of the cocktail (150 mg of Tixagevimab and 150 mg of Cilgavimab). Those who received the cocktail showed a 69% lower chance of COVID-19 symptoms or death from any cause (*p* = 0.002, 95% CI 36–85) in a median follow-up of 83 days. Only one (0.1%) of the cocktail recipients needed hospitalization, compared to 27 (0.6%) of the controls (*p* = 0.07) [67]. Some additional cohort clinical studies were conducted on organ transplant recipients wherein BA.2 was the immune strain, with all these studies reporting increased neutralization and a decreased infection rate [68,69].

Since the cocktail was developed before the prevalence of Omicron, a 150 mg dosage has been reported to be insufficient against BA.1 and BA.1.1. Hence, an increased dosage using 300 mg of Tixagevimab plus 300 mg of Cilgavimab to elicit greater neutralization is proposed and is now listed as one of the recommended treatments by the National Institutes of Health (NIH) [66,67]. However, research examining its efficacy against the current BA.4/BA.5 variants of concern is lacking, and no clinical trials have yet been conducted [66].

#### 5.2.2. Amubarvimab and Romlusevimab

The first antibody cocktail therapy to be commercially licensed in China was a cocktail composed of Amubarvimab and Romlusevimab. These two anti-COVID-19 monoclonal antibodies were obtained in early 2020 during the pandemic. According to the trial results from NIH’s ACTIV-2 randomized, blinded clinical trial, a total of 837 mild–moderate, non-hospitalized adult patients participated in the trial. In the 7-day follow-up, recipients of Amubarvimab and Romlusevimab reported fewer hospitalizations (12 vs. 45) and deaths (1 vs. 9), while the 28-day follow-up revealed a 78% decrease in both hospitalization and mortality. Additionally, fewer adverse events were reported in the cocktail-receiving group compared to the placebo group (3.8% vs. 13.4%). Due to its safe and effective treatment of COVID-19, it is now listed as one of the Emergency Use Authorizations by the FDA [70].

#### 5.2.3. Bamlanivimab and Etesevimab

The SARS-CoV-2 spike protein targets the powerful neutralizing IgG1 mAb known as Bamlanivimab (LY-CoV555). It prevents viral attachment to human cells and penetration, neutralizing the virus and perhaps preventing and curing COVID-19 [39]. According to studies, prophylactic dosages such as 2.5 mg/kg suppressed viral proliferation in the upper and lower respiratory tract in samples obtained on research day 6 following viral injection in a rhesus macaque challenge model. A range of COVID-19 indications, including prevention and therapy, are being assessed for this antibody and it has begun clinical testing [71].

Etesevimab (JS016 or LY-CoV016) is another humanized neutralizing mAb. It can selectively bind to the virus’s surface protein receptor-binding domain (RBD) and efficaciously prevent it from attaching to the ACE-2 receptors on the host cell surface [72].

Etesevimab and Bamlanivimab are frequently used in tandem. In one phase 2 randomized clinical trial involving 577 non-hospitalized patients by Gottlieb et al., individuals with mild to moderate illness receiving combination treatment of 2800 mg Bamlanivimab and 2800 mg Etesevimab showed a significantly higher reduction in virus load at day 11, compared to the placebo group. The difference in viral log load change from baseline at day 11 compared with placebo was −0.57 (95% CI, −1.00 to −0.14; *p*  =  0.01). Furthermore, with only one incidence of COVID-19-related hospitalization or emergency department visit, combination treatment showed the best performance compared to sole Bamlanivimab or placebo treatment [73].

In another phase 3 trial, 1035 mild or moderate patients randomly received a Bamlanivimab/Etesevimab combination or placebo treatment in a one-to-one ratio. At day 7, patients who were administered Bamlanivimab with Etesevimab reported a higher viral log load decrease than in the placebo group (95% CI, 1.46 to 0.94; *p* = 0.001). At day 29, 36 of 517 (7.0%) placebo recipients and 11 of 518 (2.1%) Bamlanivimab/Etesevimab recipients experienced a hospitalization or COVID-19-related death. While 10 mortality cases were recorded in the placebo group, none were recorded in the Bamlanivimab/Etesevimab group [73].

Although the combination of Bamlanivimab and Etesevimab seemed successful for mild or moderate patients in the two clinical studies mentioned above, their application was discontinued in June 2021 because this combination could no longer provide sufficient efficacy against the newly emerged Omicron variant.

#### 5.2.4. Casirivimab and Imdevimab

Casirivimab and Imdevimab are two other human monoclonal antibodies used in combination. The neutralizing cocktail composed of them is called REGN-COV2. They act to prevent virus attachment by interacting with the virus’s spike protein RBD area [74]. Weinreich et al. conducted a phase 1–3, randomized, double-blind clinical study to examine the effectiveness of REGN-COV2. A total of 275 non-hospitalized patients were randomly assigned to one of three treatment groups: placebo, low-dose (2.4 g), or high-dose REGN-COV2 (8.0 g). As a result, 6% of participants in the control group and 3% of those who received REGN-COV2 in the entire study population reported having at least one medical visit. Moreover, this REGN-COV2 combination was observed to decrease the viral load in patients, with a larger impact on individuals whose immune response had not yet begun to react or in those with high viral loads at baseline. However, the proportion of recipients who experienced adverse effects (e.g., hypersensitivity response) was comparable to that of the placebo control group [75].

As with the Bamlanivimab/Etesevimab combination, REGN-COV2 has also been considered non-therapeutic due to the current prevalence of the Omicron strain and its substrains. Therefore, it is also now listed as a treatment option that is not recommended.

#### 5.2.5. Sotrovimab

The anti-spike-neutralizing monoclonal antibody Sotrovimab was applied to treat mild to moderate COVID-19 in outpatient settings. It is resistant to all key protein mutations in Omicron mutant strains. Sotrovimab is currently approved in approximately 12 countries, including the U.S. In clinical trials, patients with mild to moderate new coronary arteries experienced a 79% reduction in the risk of hospitalization and mortality. In an interim analysis of the ongoing COMET-ICE research, the effectiveness of Sotrovimab was assessed. Patients received either a placebo (N = 292) or a single 500 mg infusion of Sotrovimab over 1 h. The whole randomized population’s median age was 53 years (range: 18–96). Compared to the placebo group, Sotrovimab patients’ clinical COVID-19 advancement was reduced by 85% on day 29 (*p* = 0.002) [76].

#### 5.2.6. Tocilizumab

Tocilizumab, a recombinant humanized IgG1 mAb, binds exclusively to membrane-bound and solubilized IL-6 receptors (sIL-6R and mIL-6R), blocking the signaling cascade and lowering IL-6’s pro-inflammatory activity. Between March and August 2021, retrospective cohort research was carried out involving persons with severe SARS-CoV-2 pneumonia. A total of 101 individuals were included, of whom 46 received corticosteroids and 55 received corticosteroids together with tocilizumab. The findings revealed a 58-year-old median age and a 63.9% female population. Obesity and high blood pressure were observed in 36.1% and 54.6%, respectively. The cohort’s survival rate was 81.4%, and the average hospital stay was 19.0 days. In the cohort, secondary infections were observed in 47.4% of cases. Patients in the tocilizumab group had shorter hospital stays, lower C reactive protein (CRP) levels at discharge, a decreased likelihood of multiple organ failure, and higher functional status. According to a bivariate study, there were no differences in mortality rate or secondary infection occurrence. There was a significant difference in the variability of the clinical state as measured by the WHO Ordinal Scale from deterioration to discharge (or 14 days), demonstrating a better functional status in patients receiving tocilizumab [77].

#### 5.2.7. Bebtelovimab

Bebtelovimab is a mAb with newly identified antiviral activity against the prevalent Omicron strain and its substrains. Although clinical data are still lacking, it has been listed as one of the recommended therapies for non-hospitalized adults with mild to moderate COVID-19 symptoms [78]. In a retrospective cohort study, despite being, on average, 10 years older than the control group, the Bebtelovimab-treated group saw fewer hospitalizations or fatalities (3.1% vs. 5.5%). In other words, Bebtelovimab can bring an even lower rate of hospitalization or death despite patients being older, in poorer health conditions, and more likely to be immunocompromised [79].

### 5.3. Other COVID-19 Treatments No Longer Used Due to Inefficacy

#### 5.3.1. Convalescent Plasma

Convalescent plasma treatment uses blood from patients who have recovered from the disease to aid the healing of others. In treating COVID-19, emergency treatment with high-antibody-titer convalescent plasma has received approval from the U.S. Food and Drug Administration (FDA). It appeared promising and was widely accepted by many clinicians internationally. However, there is insufficient support that it can substantially help in fighting against infection and optimizing the patient’s prognosis. According to three meta-analysis reports, two studies showed that convalescent plasma therapy did not have a positive impact on the disease, while the other analysis concluded that it could reduce mortality [80,81,82]. In addition, one research work (n = 511) examining the impact of convalescent plasma on the development of COVID-19 in high-risk individuals suggested that convalescent plasma had no beneficial effects on the course of the illness or the time of hospitalization [83]. The effectiveness of convalescent plasma in treating COVID-19 requires further investigation. Judging from the current results and the NIH’s treatment recommendations, it is not recommended to use convalescent plasma collected before the Omicron epidemic [66]. Moreover, it is not recommended to use convalescent plasma to treat hospitalized patients [66].

#### 5.3.2. Chloroquine

Chloroquine is a cheap drug with a long history that has recently been reported to have a broad-spectrum antiviral capability and immune-modulating activity [41,84,85]. It was originally used to treat malarial and autoimmune diseases and is currently under evaluation in an open trial (ChiCTR2000029609). Chloroquine inhibits the viral infection of the cells by escalating the endosomal PH that is essential for virus/cell fusion, and, meanwhile, it interferes with the glycosylation of the SARS viruses’ receptors [86]. A study showed that the drug functions at both the initial and post-entry of viral infection [41]. Moreover, it was reported to be widely spread in the whole body, including the lungs [41]. The EC_90_ value of chloroquine against the SARS-CoV-2 infection of E6 cells was 6.90 μM, whereas the EC_50_ value was 1.13 μM (SI  >  88.50) [41]. Thus far, more than 100 patients have taken chloroquine phosphate to treat the disease, and there have been no drug-related adverse effects. Since it has been used for over 70 years in treating other diseases, experts consider it controllable and safe to expand the treatment to a wider population [87]. Nevertheless, whether chloroquine can benefit patients remains unconfirmed, as large clinical double-blind trials have not been reported [88]. In general, some studies have shown that chloroquine is effective, while other studies have shown that chloroquine is ineffective. However, there is currently no definitive evidence that chloroquine is effective or ineffective. The latter trend is becoming increasingly apparent.

#### 5.3.3. Hydroxychloroquine

Hydroxychloroquine is a 4-aminoquinoline derivative and an antimalarial drug. Its action and mechanism are similar to those of chloroquine. However, its toxicity is only half that of chloroquine. Medical doctors from Wuhan University People’s Hospital found that all the patients in their hospital with systemic lupus erythematosus were not infected with SARS-CoV-2. Hence, they suspected that the long-term administration of hydroxychloroquine may exert a beneficial effect in treating systemic lupus erythematosus, which helps to prevent the infection. This seems a reasonable and plausible explanation since hydroxychloroquine has a similar treatment effect as chloroquine, and chloroquine, as mentioned above, is effective against SARS-CoV-2. Therefore, they treated 20 patients using hydroxychloroquine and found that the clinical symptoms improved significantly in one to two days. A review of chest CT after five days of drug use showed a significant improvement in absorption in 19 cases. The remaining patient (who previously had renal insufficiency) had progressive lesions in the chest CT. However, the patient’s clinical symptoms improved significantly on the second day of hydroxychloroquine use. In addition, none of the normal patients in the group developed severe illness, and one was discharged/released on February 13 [89]. This result preliminarily confirmed the short-term efficacy of hydroxychloroquine in the treatment of new coronary pneumonia, which can effectively relieve symptoms, reverse the rate of exacerbation, and shorten the course of the disease. However, when combining hydroxychloroquine with azithromycin to treat the disease, QTc prolongation indicated a high risk of arrhythmia. Further application trials of hydroxychloroquine in severe and critical neo-coronary pneumonia are ongoing [90].

#### 5.3.4. Ivermectin

Ivermectin is a drug widely used to treat parasitic diseases. It was not approved to treat any viral disease prior to the COVID-19 outbreak. According to in vitro research, Ivermectin can prevent the virus from using the host’s importin alpha/beta-1 nuclear transport proteins to hinder the body’s antiviral response [91,92]. Ivermectin also has anti-inflammatory properties and can prevent viruses from adhering to host cells’ surfaces, making it feasible to treat COVID-19 [20,93,94,95]. Some studies, including meta-analyses and randomized controlled clinical trials, assessed its effectiveness and safety in treating COVID-19. However, most studies did not note statistically significant variations in most parameters, including death rates, hospitalization durations, clinical endpoints, illness progression, and prognosis [96,97,98]. Hence, the use of Ivermectin in managing COVID-19 is not recommended.

#### 5.3.5. Niclosamide

Niclosamide is another drug that was originally used in the treatment of parasites (tapeworm infection). This drug has been recognized with antiviral potential as it showed antiviral activity against the influenza virus and HRV [99]. Several studies discovered niclosamide as a potential treatment candidate against COVID-19. Niclosamide’s potential mechanisms of action against COVID-19 include preventing viral entry by modifying the endosomal pH and preventing viral multiplication by inhibiting autophagy [100].

In a randomized controlled clinical study, Abdulamir et al. compared the treatment of 75 COVID-19 patients with niclosamide to 75 COVID-19 patients treated with only standard-of-care medicine. As the number of patients who died was the same in both groups, the results showed that niclosamide did not improve the survival rate of patients with severe COVID-19 (*p* > 0.05). In patients with moderate and severe COVID-19, niclosamide reduced the recovery period by roughly 5 and 3 days compared to the controls (*p* ≤ 0.05). Moreover, it can shorten the recovery time in patients with comorbidities by five days (*p* ≤ 0.05), while that in patients without comorbidities is only one day (*p* > 0.05) [96]. Thus, they concluded that it is a useful and safe adjunct therapy for the treatment of severe COVID-19 patients, particularly those with comorbidities [96].

#### 5.3.6. Chinese Traditional Medicine

Chinese traditional medicine (TCM), also known as Chinese herbs, has been reported to play a role in treating the disease [101]. According to statistics, at 00:00 on February 5, four pilot provinces used TCM to treat 214 confirmed cases and showed an over 90% rate of effectiveness, among which the symptoms of 60% of patients showed a significant improvement, and 30% of patients had stable symptoms without exacerbation [102]. Despite some successful application cases, traditional Chinese medicine preparations’ specific mechanisms and efficacy still need more investigation.

#### 5.3.7. Dietary Therapy

Supplementations such as vitamin C, vitamin D, and zinc have all been postulated to affect COVID-19 patients positively. Only one trial found that administering vitamin C supplements to patients reduced mortality and the requirement for respiratory support [103]. However, several other publications stated that little evidence shows any positive impact. Analysis of the available data demonstrates that these supplementations are all incapable of lowering the death rate, ICU admission likelihood, or need for respiratory support [104,105,106].

In addition, probiotics are another factor hypothesized to affect the recovery of COVID-19 patients. Through direct antagonism with pathogenic bacteria, probiotics can activate the intestinal immune system to prevent a viral attack, bolster the immune system, improve mucosal barrier function, and limit bacterial adhesion and invasion in the intestinal epithelium [68].

## 6. SARS-CoV-2 Vaccine Trial

As drugs are under development, vaccine development has also made progress. Thus far, the vaccines include five categories: inactivated whole virus vaccines (Sinovac’s CoronaVac and Sinopharm’s Covilo), viral vector vaccines (Covishield ChAdOx1, AstraZeneca’s Vaxzevria and CanSino’s Convidecia), live attenuated whole virus vaccines (Codagenix’s CoviLiv and Indian Immunological’s iNCOVACC), messenger RNA (mRNA) and DNA-based vaccines (Pfizer–BioNTech’s Comirnaty BNT162b2, Moderna’s Spikevax mRNA-1273, and Inovio’s INO-4800), and recombinant protein vaccines (Nuvaxovid NVX-CoV2373 and Covovax NVX-CoV2373). In addition, other categories of vaccines are being developed [107]. The World Health Organization (WHO) reported that more than 300 COVID-19 vaccines were in preclinical or clinical development as of September 2022. Some vaccines that have undergone trials for the treatment of SARS-CoV-2 are summarized below (Table 1).

With the outbreak of SARS-CoV-2, vaccination is still one effective means of preventing infection [108]. However, the protection period of the SARS-CoV-2 vaccine is uncertain [109]. The SARS-CoV-2 vaccine protection period diminishes over time, and current data suggest that, for most people, the effectiveness of the SARS-CoV-2 vaccine’s potent prevention of severe illness and death lasts for at least 5–6 months [110]. After six months of initial vaccination, a second or third vaccination is needed to improve the vaccine’s protection cycle. However, after a second or third vaccination, an individual can still be infected with the virus [110]. The highest protection is achieved several weeks after the vaccination. If the vaccine is given in two doses, complete immunity is not achieved until 2–4 weeks after the second dose. Viral infection may still occur during this period. Though the SARS-CoV-2 vaccine is highly effective in preventing severe illness and death, a 100% effective vaccine is unavailable [111].

Actual data show that the currently used vaccine against the most prevalent SARS-CoV-2 mutant strains (Alpha, Delta, Omicron) still has short-term efficient human protection. In the United Kingdom, the adequate protection against symptomatic disease was slightly lower for B.1.617.2 (Delta) than for B.1.1.7 (Alpha) in individuals receiving two doses of the BNT162b2 (Pfizer BioNTech) vaccine or the ChAdOx1-S (AstraZeneca) vaccine [112,113].

Omicron is the currently circulating variant of concern (VOCs). Omicron was first detected in South Africa on November 9, 2021. There was a significantly lower neutralizing antibody response to Omicron in vaccinated individuals compared to the original SARS-CoV-2 strain or Delta (B.1.617.2) [114].

According to an up-to-date retrospective cohort study involving 490,838 individuals, a booster of BNT162b2 or mRNA-1273 is similarly effective in death prevention and can yield up to 4 months’ moderate protection against infection or hospitalization after inoculation in the Omicron variant era. As results indicated, 42.3% of individuals who received a booster vaccine had effectiveness of longer than 10 days, while 53.3% were protected from hospitalization and 79.1% from death [115].

Another retrospective cohort study analyzed 5609 individuals taking immune suppressants and found that three doses of BNT162b2 and mRNA-1273 provide effectiveness of 50% and 60% against the Omicron variant, respectively [116]. Furthermore, three doses of BNT162b2 and mRNA-1273 can both provide effectiveness of 87% against hospitalization caused by the Omicron variant in immunosuppressed individuals [117].

Moreover, according to a study by Zhao et al., a high-dose 5 mg inactivated vaccine named KCNOVAC as a third booster injection administered to 63 volunteers could induce positive effects in boosting cellular immunity and variant-specific neutralizing antibodies against Omicron [118].

As for trials against current circulating lineages BA.4 and BA.5, the bivalent Omi-cron-containing mRNA-1273.214 booster is reported to be more effective in protecting against Omicron subvariants than mRNA-1273. Participants in a study received 50 μg of either mRNA-1273.214 or mRNA-1273 as single second booster doses. It was then discovered that the mRNA-1273.214 group produced higher titers of neutralizing antibodies against Omicron BA.1, BA.4, and BA.5 than the mRNA-1273 group [119]. A bivalent mRNA-1273.214 vaccine is a promising tool in the fight against novel variations because it elicits a greater spike-binding antibody response against Omicron and the majority of other variants than mRNA-1273 [120].

In a case–control study, the effective protection period of vaccines against virulent strains was assessed using the BNT162b2, ChAdOx1 CoV-19, and mRNA-1273 vaccines in patients, with 204,154 eligible Delta strain and 886,774 eligible Omicron strain cases [121].

The vaccine was administered as an initial and booster dose, and the results showed an effective protection rate of 65.5% at 2–4 weeks for two doses of the BNT162b2 vaccine, decreasing to 8.8% at 25 weeks or longer [122]. For the ChAdOx1 nCoV-19 vaccine in the initial and booster doses of the BNT162b2 vaccine, the vaccine efficacy rate increased to 62.4% at 2–4 weeks and decreased to 39.6% at 10 weeks or more of vaccination [123]. For the initial BNT162b2 vaccine, and among those receiving a booster of the NT162b2 vaccine, vaccine effectiveness increased to 67.2% and decreased to 45.7% at 10 weeks or longer. For the ChAdOx1 nCoV-19 vaccine initial and booster mRNA-1273 populations, vaccine protection increased to 70.1% at 2–4 weeks but decreased to 60.9% at 5–9 weeks. The BNT162b2-vaccinated and booster mRNA-1273-vaccinated populations showed improved vaccine efficacy to 73.9%, which decreased to 64.4% at 5–9 weeks. Primary immunization with two doses of ChAdOx1 nCoV-19 or BNT162b2 vaccine provided limited protection against symptomatic disease caused by the Omicron variant. After the introductory course of [123], this protection diminished over time [124].

These results indicate that receiving an additional mRNA booster can provide additional protection against COVID-19-related manifestations. Especially in the present context, whereby Omicron is the circulating variant of concern, mRNA booster vaccination is the only widely proven vaccination protocol against the Omicron variant.

### Factors Affecting Vaccine Effectiveness 

A number of elements, which are summarized in the following sections, affect the COVID-19 vaccine’s efficacy differently. The effectiveness of COVID-19 vaccination is primarily affected by the various COVID-19 infection levels in patients, the patient’s age, and the vaccine’s type and quality [91,125]. Secondly, most clinical trials have found that the vaccine can prevent and treat clinical symptoms associated with COVID-19 from the second dose onwards and is most effective after the third dose [126]. If annual injections are maintained, the vaccine’s effectiveness can be well maintained. Thirdly, improper transport and storage of vaccines can also lead to reduced efficacy or failure. For example, the mRNA lipid nanoparticle COVID-19 vaccine must be stored at low temperatures, or its unstable nature can lead to degradation [126]. Fourthly, SARS-CoV-2 is subject to frequent mutations, sometimes producing new mutant strains with greater transmission, such as Delta and Omicron, which could potentially impact vaccine efficacy [127]. The prophylactic treatment of reinfection with different strains of the COVID-19 vaccine is also a great challenge for people exposed to COVID-19 for a long time [127]. Finally, the safety and efficacy of the COVID-19 vaccine in specific populations, such as pregnant women, cancer patients, and patients with immunodeficiency diseases, remains to be determined. The different individual health conditions of specific populations are also a factor that cannot be ignored in the effectiveness of the COVID-19 vaccine [128].

The WHO target product profile for the COVID-19 vaccine states that the minimum protection required for the COVID-19 vaccine is 50.00% and that the ideal vaccine protection is 70.00% and above. The data show that BNT162b2 provides 95% protection against COVID-19 and has a low rate of adverse reactions [129]. The main adverse reactions after vaccination include, firstly, local reactions, including pain, swelling, and local itching at the injection site; and, secondly, systemic reactions, mainly fever and malaise.

Although the incidence is low, there is still the potential for severe adverse reactions. An Israeli study found a risk of myocarditis and other severe adverse reactions, including pulmonary embolism, among BNT162b2 vaccine recipients. This study also found a three-fold increase in myocarditis risk after vaccination [129].

Multiple mutations of SARS-CoV-2 have been disseminated throughout the population. Given that the transmissibility and toxicity of the mutants have been improved to varying degrees, it is crucial to determine whether the currently available vaccinations are still effective against mutations. Patients who have received vaccinations can also develop the Omicron form, resistant to most therapeutic antibodies [130].

## 7. Concluding Remarks

SARS-CoV-2, which has caused an epidemical pneumonia outbreak, is a new betacoronavirus that belongs to the subgenus Sarbecovirus [19,131,132]. As SARS-CoV-2 spreads globally, controlling its dissemination is becoming an urgent issue. It is necessary to highlight that mortality rates in older men with SARS-CoV-2 infection and other underlying diseases are usually greater than in older women or younger, potentially healthier patients [133,134,135]. More studies are warranted to identify the contributing factors responsible for this finding.

Although there have already been more than 630 million verified cases detected worldwide, the epidemic’s peak has yet to be determined. Accelerated vaccination and global equity are the top priorities. Vaccination remains the most effective strategy against outbreaks where the virus cannot be eliminated. Public health and social measures continue to be implemented. Vaccines alone will not end this outbreak, particularly given the variable vaccination rates and the potential for new mutant strains that can evade vaccines’ effectiveness. Effective preventive methods and strict public health measures are still required. Although there is currently some international cooperation in the fight against this epidemic, nations are more likely to blame one another. Humanity continues to struggle with this disease, and it should not be politicized. Prejudice has no place in our efforts to control the pandemic; only collaboration based on science and reason will enable us to alleviate the crisis and resume normal social interactions.

## Figures and Tables

**Table 1 vaccines-10-02145-t001:** Summary of some SARS-CoV-2 vaccines.

Vaccine	Platform	Manufacturer	Description
BNT162b2	Nucleoside-modified mRNA	BioNTech Manufacturing GmbH (Mainz, Germany)	Neo-coronavirus infectious disease (COVID-19) vaccine candidate BNT162b2 was created by BioNTech and Pfizer and given intramuscularly. It is composed of lipid nanoparticles and nucleoside-modified mRNA that encodes the pinched mutant protein of SARS-CoV-2.
AZD1222	Recombinant ChAdOx1 adenoviral vector encoding the spike protein, the antigen of SARS-CoV-2	AstraZeneca (Cambridge, England)	It uses a replication-deficient adenovirus that can infect chimpanzees as a vector. The genetic material of the novel coronavirus (SARS-CoV-2) stinger protein is present in it, which enables the body to produce the surface stinger protein after vaccination and develop immunity against the novel coronavirus.
ChAdOx1_nCoV-19	Recombinant ChAdOx1 adenoviral vector encoding the spike protein, the antigen of SARS-CoV-2	Serum Institute of India Pvt., Ltd. (Pune, India)	The vaccine is a chimpanzee adenovirus vector (ChAdOx1) expressing the SARS-CoV-2 spike protein.
Ad26.COV2.S	Recombinant, replication-incompetent adenovirus type 26 (Ad26) vectored vaccine encoding the SARS-CoV-2 spike (S) protein	Janssen–Cilag International NV (Beerse, Belgium)	The SARS-CoV-2 stinger protein is delivered into host cells by an adenovirus serotype 26 (Ad26), a common cold virus, and then stimulates the body to increase its immune response to COVID-19. However, thrombocytopenic thrombosis is an uncommon adverse effect of vaccination.
mRNA-1273	mRNA-based vaccine encapsulated in a lipid nanoparticle (LNP)	Moderna Biotech (Cambridge, United States)	mRNA-1273 is a messenger ribonucleic acid (mRNA) vaccine against a novel coronavirus that acts against the stinger protein of the virus.
SARS-CoV-2 Vaccine (Vero Cell)	Inactivated, produced in Vero cells	Beijing Institute of Biological Products Co., Ltd. (BIBP) (Beijing, China)	The vaccine is an inactivated viral vaccine that stimulates somatic immunity and has high vaccine protection but requires a late booster.
CoV2373	Recombinant nanoparticle prefusion spike protein formulated with Matrix-M™ adjuvant	Novavax (Gaithersburg, United States)	NVX-CoV2372 is Novavax’s vaccine candidate against SARS-CoV-2, the virus that causes the new coronavirus. The vaccine incorporates Novavax’s proprietary saponin-based Matrix-M™ adjuvant, which enhances antigen presentation in local lymph nodes by enabling the adjuvant to stimulate antigen-presenting cells to enter the injection site to enhance the immune response and produce high levels of neutralizing antibody production.
Ad5-nCoV	Recombinant novel coronavirus vaccine (adenovirus type 5 vector)	CanSinoBIO (Tianjin, China)	Ad5-nCoV is the gene for the neo-coronavirus S protein built into the adenovirus genome. The outer shell remains the normal outer shell protein of the adenovirus, but the genes inside contain the genes encoding the neo-coronavirus S protein. Therefore, when the adenovirus infects the host cell, it releases all the genes encoding the neo-coronavirus S protein into the host cell and synthesizes the S protein in the cytoplasm, which stimulates a series of immune responses.
Sputnik V	Human adenovirus vector-based COVID-19 vaccine	Russian Direct (Moscow, Russia)	The Sputnik vector vaccine is based on adenovirus DNA, in which the SARS-CoV-2 coronavirus gene is integrated. Adenovirus is a “container” to deliver the coronavirus gene to cells and synthesizes the SARS-CoV-2 virus’s envelope proteins, “introducing” the immune system to a potential enemy.
SCB-2019	Novel recombinant SARS-CoV-2 spike (S)–trimer fusion protein	Clover Biopharmaceuticals (Shanghai, China)	SCB-2019 is a protein subunit vaccine candidate containing a stable trimeric form of the spike (S) protein (S-trimer) and two different adjuvants.

## Data Availability

Not applicable.

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
