# Peer review of "SARS-CoV-2: An Updated Review Highlighting Its Evolution and Treatments"

_vaccines, 2022, doi:10.3390/vaccines10122145_

Round 1

Reviewer 1 Report (Previous Reviewer 2)

Dear Authors,

I have read your manuscript carefully. Several issues are present. In particular, your revision is too superficial because it merely describes a small part of the scientific literature available for these arguments.

Regarding symptoms, I do not understand what reference (19) you choose (the link is not working. I suggest you cite scientific publications (e.g. https://doi.org/10.1371/journal.pone.0248009, 10.1080/17512433.2021.1902303). In addition, two important symptoms are missing. 1) skin manifestation (e.g. https://doi.org/10.1097/IPC.0000000000000952, 10.1016/S1473-3099(20)30402-3); 2) anosmia and ageusia (e.g. 10.1002/hed.26204, 10.1002/alr.22995);

About the sentence, "Only a minority of the patients in the present cases have a critical condition, giving the majority of patients a decent outlook. Older people and those with persistent underlying conditions have a dismal prognosis." references are missing.

About the sentence "Children's symptoms are generally modest", you still use the reference number 19. As mentioned above, I suggest you cite scientific papers (e.g. 10.26355/eurrev_202004_21043).

You wrote, "hildren's symptoms are generally modest [19]. The pathological features of COVID-19 are very similar to those of SARS and MERS infections [28]. Additionally, liver biopsy samples from COVID-19 patients revealed mild active inflammation and moderate micro-vascular septoplasty in the hepatic lobular manifold region, indicating that the injury may result from a drug-induced liver injury or a SARS-CoV-2 infection. The cardiac interstitium had a little inflammatory infiltrate of mononuclear cells. However, no more myocardial parenchymal damage was identified [28]. CD4+ and CD8+ cell accounts in peripheral blood were greatly reduced, but they were over-activated. In addition, high pro-inflammatory CCR4+ CCR6+ Th17 cells increased. CD8+ T cells had higher cytotoxic granules [28]. A study showed that the chest radiograph rapidly progressed pneumonia in both lungs [28]." For all of this important information, you cite a case report from February 2020. I think that you should carefully revise the references you use. A review should report the last and most valuable information available.

Paragraph 5, about treatments available, is still very lacking. 

Firstly the division between table 1 and table 2 is non-sense. Why do you put in table 1 many types of drugs and in table 2 only monoclonal antibodies?
I suggest you divide for the time of use. For example, molnupiravir and sotrovimab are treatments administered to people without oxygen supplementation to avoid disease progression.
Also, remdesivir (3-days) is administered to avoid disease progression. On the contrary remdesivir 5-days course is indicated for people who need oxygen supplementation to reduce death.

In addition, I suggest describing these treatments better, using all information available. For example, about remdesivir, there are many studies with controversial results; Therefore, I recommend using more than one study to describe its use.

Furthermore, it would be best to describe all treatments available and approved by international guidelines. e.g. nirmatrelvir/r is not present in your revision; tocilizumab is available in two different formulations: intervenous or subcutaneous. 

Regarding the vaccines, I suggest adding a table.

Author Response

Point 1: Regarding symptoms, I do not understand what reference (19) you choose (the link is not working. I suggest you cite scientific publications (e.g. https://doi.org/10.1371/journal.pone.0248009, 10.1080/17512433.2021.1902303). In addition, two important symptoms are missing. 1) skin manifestation (e.g. https://doi.org/10.1097/IPC.0000000000000952, 10.1016/S1473-3099(20)30402-3); 2) anosmia  (e.g. 10.1002/hed.26204, 10.1002/alr.22995);

Response: We are very grateful that Reviewer #1 gave us pertinent suggestions to improve the manuscript. We carefully revised the references and added skin manifestation and anosmia symptoms to the manuscript

Point 2: About the sentence, "Only a minority of the patients in the present cases have a critical condition, giving the majority of patients a decent outlook. Older people and those with persistent underlying conditions have a dismal prognosis." references are missing.

About the sentence "Children's symptoms are generally modest", you still use the reference number 19. As mentioned above, I suggest you cite scientific papers (e.g. 10.26355/eurrev_202004_21043).

Response: Thanks for your advice. We have carefully revised the references and rewrote the symptoms for the elderly and children.

We had added: "The most prevalent clinical signs are fever, dry cough, and malaise. A few people experience diarrhea, myalgia, conjunctivitis, runny nose, and nasal congestion [89]. Children often experience minor symptoms within a week, with mild to moderate fever as the most common symptom. A few children can experience high fever, dry cough, and other respiratory symptoms". 

Point 3: You wrote, "hildren's symptoms are generally modest [19]. The pathological features of COVID-19 are very similar to those of SARS and MERS infections [28]. Additionally, liver biopsy samples from COVID-19 patients revealed mild active inflammation and moderate micro-vascular septoplasty in the hepatic lobular manifold region, indicating that the injury may result from a drug-induced liver injury or a SARS-CoV-2 infection. The cardiac interstitium had a little inflammatory infiltrate of mononuclear cells. However, no more myocardial parenchymal damage was identified [28]. CD4+ and CD8+ cell accounts in peripheral blood were greatly reduced, but they were over-activated. In addition, high pro-inflammatory CCR4+ CCR6+ Th17 cells increased. CD8+ T cells had higher cytotoxic granules [28]. A study showed that the chest radiograph rapidly progressed pneumonia in both lungs [28]." For all of this important information, you cite a case report from February 2020. I think that you should carefully revise the references you use. A review should report the last and most valuable information available.

Response: Thank you for your suggestion. We have rewritten it in the manuscript.

Point 4: Firstly the division between table 1 and table 2 is non-sense. Why do you put in table 1 many types of drugs and in table 2 only monoclonal antibodies?

I suggest you divide for the time of use. For example, molnupiravir and sotrovimab are treatments administered to people without oxygen supplementation to avoid disease progression.

Response: Thank you for your suggestion. We merged the contents of Table 1 and Table 2 into the body and deleted the table.

Point 5: In addition, I suggest describing these treatments better, using all information available. For example, about remdesivir, there are many studies with controversial results; Therefore, I recommend using more than one study to describe its use.

Response: Thank you for your suggestion. We have added some content about remdesivir.

Point 6: Furthermore, it would be best to describe all treatments available and approved by international guidelines. e.g. nirmatrelvir/r is not present in your revision; tocilizumab is available in two different formulations: intervenous or subcutaneous.

Response: Thank you for your suggestion. More treatment options are now added to our manuscript under point 5 ("5. Treatments under development"). All changes can be seen under track change mode.

Point 7: Regarding the vaccines, I suggest adding a table.

Response: Thank you for your suggestion. We have added about the current mainstream vaccines worldwide and made Table 1.

Reviewer 2 Report (Previous Reviewer 1)

The review article by Zhang et al. has summarized the updated literature highlighting the SARS-COV-2 evolution and treatment options against it. However, a few concerns are suggested below to enhance the paper's quality.

  1. In the abstract, the authors have mentioned only plasma and monoclonal antibodies, however, a large section is dedicated to drugs. Please improve the abstract accordingly.
  2. In table 2, please add an introduction line for Sotrovimab first as the authors have mentioned for others.
  3. Please explain what authors mean when they refer to “Light individuals” in the sentence “Light individuals do not exhibit any symptoms of pneumonia, merely low fever, and mild weariness.
  4. Authors have mentioned “so far, the vaccines involve five categories: inactivated virus vaccine, adenovirus vector vaccine, mRNA vaccine, and recombinant protein vaccine. Which are only four, where is the fifth category of the vaccine?
  5. Authors mentioned, “Participants in a study received 50 g of either mRNA-1273.214 or mRNA-1273 as single-second booster doses.” In this, 50 g seems a very high amount of booster dose. Please be careful and re-confirm it.
  6. In the sentence “According to studies, 42,473 (43.9%) of 96,859 COVID-19 patients received at least one dosage of remdesivir. 18,328 patients (43.2%) had matches.” The authors should make it more clear what they are trying to say.

Author Response

Point 1: In the abstract, the authors have mentioned only plasma and monoclonal antibodies, however, a large section is dedicated to drugs. Please improve the abstract accordingly.

Response: We are very grateful that Reviewer #2 gave us pertinent suggestions to improve the abstract. We have now rephrased some sentences to cover the main idea of our manuscript. Changes can be seen under track change mode.

Point 2: In table 2, please add an introduction line for Sotrovimab first as the authors have mentioned for others.

Response: We thank Reviewer #2 for this question, and it is our negligence for not stating this clearly. A sentence introducing Sotrovimab has been added to the manuscript under track change mode.

We had added: "The anti-spike-neutralizing monoclonal antibody sotrovimab was used to treat mild to moderate COVID-19 in outpatient settings."

Point 3: Please explain what authors mean when they refer to "Light individuals" in the sentence "Light individuals do not exhibit any symptoms of pneumonia, merely low fever, and mild weariness.

Response: We thank the suggestion from Reviewer #2. We have now removed the term "Light individuals" and rephrased the sentence to clarify it. The change can be seen under track change mode.

We rephrased it: "Whereas people with mild symptoms will exhibit a low fever and slight fatigue instead of any pneumonia symptoms."

 Point 4: Authors have mentioned "so far, the vaccines involve five categories: inactivated virus vaccine, adenovirus vector vaccine, mRNA vaccine, and recombinant protein vaccine. Which are only four, where is the fifth category of the vaccine?

Response: We thank Reviewer#2 for pointing out this mistake. And yes, there are five types of vaccines. They are Inactivated Whole Virus vaccines, Viral Vector vaccines, Live attenuated Whole Virus vaccines, messenger RNA (mRNA), DNA-based vaccines, and Recombinant Protein vaccines. We have corrected the information, and the changes can now be seen under track change mode. 

 Point 5: Authors mentioned, "Participants in a study received 50 g of either mRNA-1273.214 or mRNA-1273 as single-second booster doses." In this, 50 g seems a very high amount of booster dose. Please be careful and re-confirm it.

Response: We thank Reviewer#2 for pointing out this mistake. We have re-confirmed it is "50 μg" instead of "50 g". The change can be seen under track change mode.

 Point 6: In the sentence "According to studies, 42,473 (43.9%) of 96,859 COVID-19 patients received at least one dosage of remdesivir. 18,328 patients (43.2%) had matches." The authors should make it more clear what they are trying to say.

Response: We thank Reviewer #3 for this question, and it is our negligence for not stating this clearly. The sentences are rephrased and can be seen under track change mode.

We had rephrased it: "According to a study, 42,473 (43.9%) of 96,859 collected COVID-19 patients received at least one dosage of remdesivir. Utilizing time-dependent propensity scores, 18,328 (43.2%) Remdesivir recipients were matched to controls."

Reviewer 3 Report (Previous Reviewer 4)

As known, there are multiple mutations of SARS-CoV-2  throughout the population. Through this paper, the authors tried to investigate the effectiveness of COVID medicines and vaccines. However, they underlined the importance of preventive measures such as social distancing to preserve public health and the need for global scientific and rational cooperation based on WHO instructions to face the pandemic. The importance of vaccination is stated. They also gave detailed information on the current medicines used in disease treatment. It is a well-written article providing more light on COVID eradication.

It is a well-written paper that could be a reference to the developed therapies and vaccines and their efficacy based on global published data.

My suggestion is to ACCEPT and publish the paper in its present form.

Author Response

Response: Thank you for your excellent comments.

Round 2

Reviewer 1 Report (Previous Reviewer 2)

Dear authors,

I have reread your manuscript. However, some issues are still present.

Regarding the clinical presentation, I suggest you add more references about it, maybe using those I have indicated in the previous review.

About remdesivir, you should also discuss some real-life data such as https://doi.org/10.1080/03007995.2022.2129801, 10.1093/cid/ciaa1041.

In addition, remdesivir (3 days course instead of 5) could also use for people with mild disease to prevent disease progression. You should read the PINETREE trial and add a sentence about it.

Please, avoid the use of commercial names (e.g. Paxlovid). 

About Nirmatrelvir/ritonavir, you should also report real-life data since the trial enrolled people with a low comorbidity burn. Therefore, I suggest you read these papers and write about them in your essay: https://doi.org/10.1002/jmv.28011, https://doi.org/10.3390/vaccines10101731, 10.1016/S1473-3099(22)00507-2.

I suggest modifying "Other potential treatment options for COVID-19" with "Other COVID-19 treatment not more used for futility" or something like that.

Author Response

Responses to comments and suggestions of Reviewer #1

Point 1: Regarding the clinical presentation, I suggest you add more references about it, maybe using those I have indicated in the previous review.

Response: Thanks for your advice. We have made reference to the previous review. and added "The death rate for COVID-19 in older patients may be higher. Older people and those with persistent underlying conditions have a dismal prognosis, the most prevalent clinical signs are fever, dry cough, and malaise. A few people experience diarrhea, myalgia, conjunctivitis, runny nose, and nasal congestion. In addition, elderly patients 65 and older are more likely to develop COVID-19 infection sequelae, including hypertension, diabetes, chronic obstructive pulmonary disease, and dysphrenia."

Point 2: About remdesivir, you should also discuss some real-life data such as https://doi.org/10.1080/03007995.2022.2129801, 10.1093/cid/ciaa1041. In addition, remdesivir (3 days course instead of 5) could also use for people with mild disease to prevent disease progression. You should read the PINETREE trial and add a sentence about it.

Response: Thank you for your suggestion. We have added some data according to your recommendation.

We had added "According to a study, Remdesivir treatment reduces mortality in SARS-CoV-2 infected individuals. In this 342 patient research, raltegravir was given to 60 patients in the control group and 18 patients in the case group (35.1% vs. 10.5%, P < .0001). Remdesivir therapy can decreased SARS-CoV-2 infected mortality. A randomized, double-blind, placebo-controlled trial validating the safety of Remdesivir in the three-day treatment. Some studies have confirmed that A 3-day outpatient course of Remdesivir significantly reduces the risk of hospitalization or death, Retrospective matched-pair study to evaluate the contribution of Remdesivir and vaccination in reducing severe clinical outcomes Patients who got a 3-day course of Remdesivir after receiving an ambulatory vaccination were 75% less likely to end up in the hospital and 95% less likely to experience respiratory failure. Neither intubations nor fatalities occurred."

Point 3: Please, avoid the use of commercial names (e.g. Paxlovid).

Response: We thank the reviewer for the reminder. And we now deleted commercials names such as "Paxlovid” and “Evusheld”.

Point 4: About Nirmatrelvir/ritonavir, you should also report real-life data since the trial enrolled people with a low comorbidity burn. Therefore, I suggest you read these papers and write about them in your essay: https://doi.org/10.1002/jmv.28011, https://doi.org/10.3390/vaccines10101731, 10.1016/S1473-3099(22)00507-2.

Response: We thank the reviewer for the suggestion that can substantially enhance our manuscript. And we have now read these real-world retrospective studies and summarized their findings in the "5.1.8 Nirmatrelvir/Ritonavir" part of our manuscript. Change can be seen under track change mode.

We had added " Nirmatrelvir/Ritonavir is another oral antiviral (OA) medication that helps prevent mild to moderate COVID patients from developing severe COVID. Naimatevir is an oral protease inhibitor that can help block the replication of the virus. Ritonavir is used as a booster for the former, slowing down the rate of decomposition and metabolism of Naimatevir in the liver to maintain the concentration of the Nirmatrelvir in the plasma and make it more effective.

To date, some real-world investigations have shown that the Nirmatrelvir/Ritonavir combination is effective against mutant strains. It is thought to be a therapy that will alter the present outbreak prevention and control strategy. One retrospective cohort study assessed the clinical outcomes of nirmatrelvir-ritonavir in mild-to-moderate hospitalized patients. Using propensity-score matching, 890 recipients of nirmatrelvir and ritonavir were matched with 890 placebo recipients. When compared to controls, beneficiaries of nirmatrelvir/ritonavir reportedly had a reduced incidence of all-cause death (P <0·0001), progression of composite disease (P <0·0001), and the requirement for oxygen therapy (P =0·032). Furthermore, a quicker reduction of viral burden in nirmatrelvir/ritonavir recipients was observed.

However, some are concerned that the addition of ritonavir, a CYP3A4 enzyme inhibitor, leads to interactions with various medications and impairs liver and kidney function, thereby restricting its usage among the general population and raising the risk of harm in the elderly and those with underlying conditions. In another real-world retrospective research by Gentile et al., 111 mild-to-moderate patients who were older or had significant comorbidities were enrolled and received the nirmatrelvir/ritonavir combination. Despite the age and comorbidities, throughout a 14-day follow-up, only one patient was reported hospitalized. The proportion of reported ADRs was low, and no recipients stopped their therapy due to adverse drug reactions. Moreover, participants with two or more comorbidities experienced similar risks of hospitalization (1.4%) versus those with none or one comorbidity (1.6%). Although they concluded that obesity was the most prevalent comorbidity for the development of severe COVID-19, these old-aged, OAs-treated recipients with comorbidities are shown to have a low hospitalization rate, mortality, and ADRs.

In short, oral antiviral medications nirmatrelvir/ritonavir demonstrated significant therapeutic benefits. The World Health Organization (WHO) and National Institute of Health (NIH) have already announced on their official websites that Nirmatrelvir, in combination with Ritonavir, may be used in mild to moderate patients who are at high risk for developing severe COVID-19."

I suggest modifying "Other potential treatment options for COVID-19" with "Other COVID-19 treatment not more used for futility" or something like that.

Response: Thank you for your suggestion. We have changed to "Other COVID-19 treatment not more used for futility."

Round 3

Reviewer 1 Report (Previous Reviewer 2)

Dear authors,

Thank you for your reply.

Re-reading your manuscript, I have only two more suggestions.

Firstly, about monlupiravir, I suggest adding more information about its use in a real-life setting. In the trial, the population was relatively young with a low comorbidity burden. In real-life studies, the population is quite different. Therefore, I suggest reading and citing these papers: https://doi.org/10.1002/jmv.28011 and https://doi.org/10.1093/cid/ciac781. In addition, this study https://doi.org/10.1002/jmv.28011 showed that people treated in the first three days since the onset of the symptoms had a lower risk of developing severe disease, and I believe it is essential information.

Secondly, I noticed that some citations should be corrected, mainly the number 30 (the author's last name is De Vito, not De) and the numbers 50 and 54 (no surnames are present, only the initials).

Author Response

Point 1: Firstly, about monlupiravir, I suggest adding more information about its use in a real-life setting. In the trial, the population was relatively young with a low comorbidity burden. In real-life studies, the population is quite different. Therefore, I suggest reading and citing these papers: https://doi.org/10.1002/jmv.28011 and https://doi.org/10.1093/cid/ciac781.

In addition, this study https://doi.org/10.1002/jmv.28011 showed that people treated in the first three days since the onset of the symptoms had a lower risk of developing severe disease, and I believe it is essential information.

Response: Thank you for your suggestion. We have added some data according to your recommendation and cited references.

We added " In a propensity score matching analysis, the risk of the composite outcome was non-significantly lower when using Molnupiravir: hazard ratio, 0.83 (95% confidence interval, 0.57-1.21). Molnupiravir was correlated with a substantial reduction in the risk of the compound outcome in elderly patients 0.54 (0.34-0.86), in women 0.41 (0.22-0.77), and in patients with insufficient COVID-19 immunization 0.45 (0.25-0.82), according to subgroup analysis. Statistics indicate that Molnupiravir may reduce the risk of severe COVID-19 and COVID-19-related death [61]. Moreover, it was reported that complications are factors in the effectiveness of Molnupiravir treatment. The mean age of the 192 patients on Molnupiravir was 70.4 ± 15.4 years, and regarding comorbidities, the most prevalent were active cancer (51, 26.6%), obesity (51, 26.6%), chronic lung disease (56, 29.2%), and cardiovascular disease (96, 50.0%). Overall, 13 patients died, six died due to COVID-19, and seven died due to complications during treatment. Notably, of these patients, one had sepsis of urinary tract infection, two had traumatic brain injuries, one had heart failure, and four had metastatic cancer [62]. In addition, People treated in the first three days after the presence of COVID-19 clinical symptoms have a lower risk of developing severe illness [62]."

References

61 Najjar-Debbiny, R.; Gronich, N.; Weber, G.; Khoury, J.; Amar, M.; Stein,     N.;Goldstein, L.H.; Saliba, W. Effectiveness of Molnupiravir in High Risk Patients: a Propensity Score Matched Analysis. Clin. Infect. Dis 2022, ciac781. [Online ahead of print]

  1. De Vito, A.; Colpani, A.; Bitti, A.; Zauli, B.; Meloni, M.C.; Fois, M.; Denti, L.; Bacciu, S.; Marcia, C.; et al. Safety and efficacy of molnupiravir in SARS-CoV-2-infected patients: A real-life experience. J. Med. Virol2022, 94, 5582-5588.

Point 2: Secondly, I noticed that some citations should be corrected, mainly the number 30 (the author's last name is De Vito, not De) and the numbers 50 and 54 (no surnames are present, only the initials).

Response: Thank you very much for your reminder, and we have revised it in the reference section.

This manuscript is a resubmission of an earlier submission. The following is a list of the peer review reports and author responses from that submission.

Round 1

Reviewer 1 Report

The review article by Zhang et al. highlights the SARS-CoV-2 treatment options and evolution findings. The authors give a brief overview of various drugs for treating SARS-CoV-2 infection, however, the section on the vaccine is insufficient. Additionally, there are some major concerns that are mentioned and suggested below for enhancing the paper's quality and visibility.

  1. The article's origin and evolution sections stand distinct from the rest of it. I suggest creating continuity and flow throughout the article.
  2. The authors should discuss more on the COVID vaccines, other treatment alternatives, and their related studies. It appears that the authors have emphasized drug treatment more and undermined the importance of vaccines.
  3. In Table 2, the authors only describe drugs based on computer computation screening that does not add any values. In my opinion, the drugs that are being extensively investigated in preclinical (in vitro and in vivo) studies should be included in this table.
  4. Conclusions should be more informative and scientific.
  5. In Table 1, finding the result information related to a drug is difficult. I recommend improving this table's formatting.

Author Response

1.The article's origin and evolution sections stand distinct from the rest of it. I suggest creating continuity and flow throughout the article.

Response: We thanks Reviewer #1 for pertinent suggestions and thoughtful comments. We have modified the order of Origin and evolution sections to ensure logical relationships in the manuscripts.

2.The authors should discuss more on the COVID vaccines, other treatment alternatives, and their related studies. It appears that the authors have emphasized drug treatment more and undermined the importance of vaccines.

Response: Thank you very much for the reminder. We have modified and added the latest research about COVID-19 vaccines.

3. In Table 2, the authors only describe drugs based on computer computation screening that does not add any values. In my opinion, the drugs that are being extensively investigated in preclinical (in vitro and in vivo) studies should be included in this table.

Response: Thank you very much for the suggestion. We modified the content in Tables 1 and Table 2, and we have added more drug and monoclonal antibody research information.

4. Conclusions should be more informative and scientific.

Response: Thank you very much for the suggestion. We have modified the conclusion section.

5. In Table 1, finding the result information related to a drug is difficult. I recommend improving this table's formatting.

Response: Thank you very much for the reminder. We have modified Table 1 and added more drug information.

Reviewer 2 Report

The manuscript does not respect the scope of the journal and the special issue. In 13 pages, only 10 lines are about vaccines. In addition, the article seems to be written 2 years ago. For example, the authors wrote "As the drugs are under development, vaccine development has also made progress [...]  Phase II and phase I trial was and will be initiated in China and America".  They have not written a single sentence about the many monoclonal antibodies we used in the last 18 months (casirivimab/imdevimab, sotrovimab), and antiviral such as molnupiravir and nirmatrelvir/r.

Author Response

The manuscript does not respect the scope of the journal and the special issue. In 13 pages, only 10 lines are about vaccines. In addition, the article seems to be written 2 years ago. For example, the authors wrote "As the drugs are under development, vaccine development has also made progress [...] Phase II and phase I trial was and will be initiated in China and America". They have not written a single sentence about the many monoclonal antibodies we used in the last 18 months (casirivimab/imdevimab, sotrovimab), and antiviral such as molnupiravir and nirmatrelvir/r.

Response: Thank you very much for the reminder. We added more research on COVID-19 vaccines, drugs, and monoclonal antibodies. The content in the manuscript has been modified and updated.

Reviewer 3 Report

The abstract appears rather outdated now, as does much of the review itself. Please see my comments in the original text.

Insufficient evidence is presented in the text to confirm or deny the possible Wuhan link with emergence of the virus.

Sections on treatments and vaccination, seem somewhat outdated now, and the conclusions unfortunately are rather dated now.

Author Response

1. The abstract appears rather outdated now, as does much of the review itself. Please see my comments in the original text.

Response: Thank you very much for the suggestions and comments in our original text. We have modified and updated the content of the abstract section in the text accordingly. 

2. Insufficient evidence is presented in the text to confirm or deny the possible Wuhan link with emergence of the virus.

Response: Thanks for your reminder. We have removed incorrect content about viral spread, and the article's content has been modified and updated.

3. Sections on treatments and vaccination, seem somewhat outdated now, and the conclusions unfortunately are rather dated now.

Response: Thank you very much for the suggestion. We have rewritten the section on vaccines and treatments in the manuscript and updated the outdated content.

Reviewer 4 Report

It is an interesting review article that collects together late information on the SARS-CoV-2 virus trying to explain the antiviral mechanism of used medicines and Chinese traditional medicines. However, the role of probiotics as a potential therapeutic approach for COVID-19 is missing (see article Stavropoulou E, Bezirtzoglou E. Probiotics as a Weapon in the Fight Against COVID-19. Front Nutr. 2020 Dec 15;7:614986. doi: 10.3389/fnut.2020.614986. PMID: 33385008; PMCID: PMC7769760)

The authors have to revise thoroughly their article , aiding more bibliography and check cautiously the English language and grammar.

The revised article form could be accepted for publication.

Author Response

It is an interesting review article that collects together late information on the SARS-CoV-2 virus trying to explain the antiviral mechanism of used medicines and Chinese traditional medicines. However, the role of probiotics as a potential therapeutic approach for COVID-19 is missing (see article Stavropoulou E, Bezirtzoglou E. Probiotics as a Weapon in the Fight Against COVID-19. Front Nutr. 2020 Dec 15;7:614986. doi: 10.3389/fnut.2020.614986. PMID: 33385008; PMCID: PMC7769760)

Response: Thank you for your suggestions. We have added probiotics and antiviral information to the manuscript. "In addition, Probiotics, through direct antagonism with pathogenic bacteria, strengthen the immune system, enhance mucosal barrier function and inhibit bacterial adhesion and invasion in the intestinal epithelium, activate the intestinal immune system to inhibit viral attack".

Round 2

Reviewer 1 Report

The review article by Zhang et al. has tried to summarize up-to-date literature focusing on SARS-CoV-2 treatment options and evolution. However, a few concerns are mentioned and proposed below for improving the paper’s quality and visibility.

  1. In Tables 1 and 2, the authors have just mentioned drugs and monoclonal antibodies and their generic mechanism of action without giving critical information such as study type, details of trials, and outcomes. Authors need to add that information to make it more impactful.
  2. This article still lacks updated information in detail. The authors have discussed only a few here and there, however, up-to-date information about the vaccines including the clinical trials for new variants should be included.
  3. Authors must improve the manuscript's flow and connections between origin, evolution, and treatment options.

Author Response

General response: We are very grateful to the reviewer for allowing us to improve further and revise the manuscript. The yellow part is the part modified after their first comments. For the second revision, any revisions for the second time suggestions to the manuscript have been marked up using the "Track Changes" in the text accordingly so that the reviewer can see the revision position of the manuscript.

Point 1: In Tables 1 and 2, the authors have just mentioned drugs and monoclonal antibodies and their generic mechanism of action without giving critical information such as study type, details of trials, and outcomes. Authors need to add that information to make it more impactful.

Response: We thanks Reviewer #1 for pertinent suggestions and thoughtful comments. We add more research status and results about drugs and monoclonal antibodies in Tables 1 and 2.

Point 2: This article still lacks updated information in detail. The authors have discussed only a few here and there, however, up-to-date information about the vaccines including the clinical trials for new variants should be included.

Response: Thank you very much for the reminder. Accordingly, we have modified and added the latest research about COVID-19 vaccines in the text.

Point 3: Authors must improve the manuscript's flow and connections between origin, evolution, and treatment options.

Response: Thank you very much for the suggestion. We have improved this section and condensed the content.

Reviewer 3 Report

Key points addressed

Author Response

Response: Thank you for your comments and suggestions on the manuscript.

Reviewer 4 Report

As mentioned, it is an interesting review article that collects together late information on the SARS-CoV-2 virus trying to explain the antiviral mechanism of used medicines and Chinese traditional medicines. The authors improved their manuscript in this last version and respected comments.

My suggestion is to ACCEPT and publish in this present form

Author Response

As mentioned, it is an interesting review article that collects together late information on the SARS-CoV-2 virus trying to explain the antiviral mechanism of used medicines and Chinese traditional medicines. The authors improved their manuscript in this last version and respected comments.

Response: Thank you for your valuable advice on the manuscript.

Round 3

Reviewer 1 Report

The authors have improved the article overall and made the necessary modifications.